# Early Fever in Allogeneic Stem Cell Transplantation: Are Presepsin and YKL-40 Valuable Diagnostic Tools?

**DOI:** 10.3390/jcm13195991

**Published:** 2024-10-08

**Authors:** Jakša Babel, Iva Košuta, Ana Vujaklija Brajković, Ana Lončar Vrančić, Vedran Premužić, Dunja Rogić, Nadira Duraković

**Affiliations:** 1Division of Intensive Care Medicine, Department of Internal Medicine, University Hospital Center Zagreb, 10000 Zagreb, Croatia; iva.kosuta@kbc-zagreb.hr (I.K.); ana.vujaklija.brajkovic@kbc-zagreb.hr (A.V.B.); 2School of Medicine, University of Zagreb, 10000 Zagreb, Croatia; nadira.durakovic@mef.hr; 3Department of Laboratory Diagnostics, University Hospital Center Zagreb, 10000 Zagreb, Croatia; ana.loncar.vrancic@kbc-zagreb.hr (A.L.V.); predstojnik.lab@kbc-zagreb.hr (D.R.); 4Division of Nephrology, Hypertension, Dialysis and Transplantation, Department of Internal Medicine, University Hospital Center Zagreb, 10000 Zagreb, Croatia; vedran.premuzic@kbc-zagreb.hr; 5Faculty of Pharmacy and Biochemistry, University of Zagreb, 10000 Zagreb, Croatia; 6Division of Hematology, Department of Internal Medicine, University Hospital Center Zagreb, 10000 Zagreb, Croatia

**Keywords:** bone marrow transplantation, hematopoietic stem cell transplantation, chitinase-3-like protein 1, YKL-40 protein, human, presepsin protein, human, procalcitonin

## Abstract

**Background**: Allogeneic hematopoietic stem cell transplantation (allo-HSCT) is a lifesaving treatment but carries a high infection risk. Diagnosing infections remains challenging due to the limited accuracy of standard biomarkers. **Methods**: This single-center study aimed to evaluate presepsin (PSP) and YKL-40 as infection biomarkers in febrile patients during the allo-HSCT pre-engraftment phase. Biomarker levels were prospectively measured in 61 febrile episodes from 54 allo-HSCT patients at admission, representing baseline levels, and then at Day 1, 3, 5, and 7 following fever onset. The diagnostic value was compared to that of procalcitonin (PCT). **Results**: PSP showed fair diagnostic value on Day 1 (AUC 0.656; 95% CI: 0.510–0.802) and Day 3 (AUC 0.698; 95% CI: 0.559–0.837). YKL-40 did not provide any significant diagnostic value across measured time points. PCT outperformed PSP and YKL-40, particularly on Day 3 (AUC 0.712; 95% CI: 0.572–0.852). When combining biomarkers, the best model for predicting infection used PSP > 3.144 ng/mL and PCT > 0.28 μg/L on Day 3, resulting in R^2^ of about 31% (*p* < 0.001). **Conclusions**: Neither test showed sufficient discriminative power for early infection to recommend their use as individual diagnostic tools in clinical practice.

## 1. Introduction

Allogeneic hematopoietic stem cell transplantation (HSCT) is a standard treatment for various malignant and non-malignant hematologic disorders. Despite notable advancements over the last decades, it is still associated with a substantial risk for infections, contributing to increased morbidity and non-relapse mortality [1,2,3]. The diagnosis of infections in everyday clinical practice remains challenging. Fever is the main clinical sign attributable to infections but is a very unspecific one, as it can also occur with non-infectious complications such as in a relapse of an underlying disease, acute graft versus host disease, and early complications of endothelial origin [4]. Failure to promptly identify infections can result in delayed antibiotic administration, which has been associated with increased mortality [5]. On the other hand, the inability to differentiate infectious from non-infectious causes of fever can lead to unnecessary empiric antibiotic treatment for many febrile patients. Hence, more specific diagnostic tools are needed to distinguish infectious from non-infectious transplant-related complications more quickly and accurately.

Presepsin (PSP) is a fragment of the monocyte/macrophage membrane cluster of the differentiation 14 (CD14) receptor [6]. The role of the CD14 receptor is to recognize and bind a variety of bacterial products, including lipopolysaccharide (LPS) and lipopolysaccharide-binding protein (LBP) complex, which is formed during systemic infections [7]. Upon binding the LPS/LBP complex to the CD14 receptor, the intracellular pro-inflammatory signaling pathway activates, leading to cytokine release and the further activation of acquired immunity [8]. A portion of the CD14 molecule is then released from the cell membrane into circulation in its soluble form (sCD14), where it is subsequently cleaved by proteases into a smaller sCD14 subtype (sCD14-ST), known as PSP [6,9]. Higher PSP levels have been observed more frequently in bacterial infections, particularly those caused by Gram-negative organisms, compared to Gram-positive and non-infective inflammatory response causes [10,11,12,13,14,15]. It has been shown that PSP could be useful in the early diagnosis, risk stratification, and prognosis of septic patients in the general population [16,17,18].

YKL-40, also known as chitinase-3-like-1 protein (CHI3L1), is a human glycoprotein, a member of mammalian chitinase-like proteins [19]. Its physiological role is still unclear, but its molecular structure suggests a potential capacity for activating innate immune response to infection [20]. Similar to PSP, studies have shown enhanced YKL-40 synthesis in different bacterial infections [21,22,23,24] and other inflammatory conditions [25,26]. Proteomic analysis identified YKL-40 as a potential biomarker of sepsis [27]. Furthermore, a fivefold increase in YKL-40 plasma concentration was demonstrated in healthy volunteers within 24 h following the injection of *Escherichia coli* endotoxin, suggesting that YKL-40 is a true acute phase reactant [28].

Thus, both PSP and YKL-40 serum levels can be regarded as indicators of activated innate immunity effector cells in response to invasive pathogens [29,30]. Their role in diagnosing infections in adult hematologic patients is poorly understood since data are scarce. Only a few studies have been conducted, mainly focusing on PSP and non-HSCT patients, with variable results [31,32,33,34,35]. As for patients undergoing HSCT, to our knowledge, only one study has evaluated the diagnostic significance of PSP as a marker of infection, with no studies carried out on YKL-40 [36]. Thus, we evaluated PSP and YKL-40 for detecting early infectious complications in febrile patients during the pre-engraftment phase of allogeneic HSCT and compared their efficacy with PCT as a standard infection biomarker.

## 2. Materials and Methods

### 2.1. Study Patients and Design

This prospective single-center observational study was conducted at University Hospital Center Zagreb, Croatia, between September 2020 and January 2023. The study was approved by the hospital ethics committee (Approval 8.1-19/285-2, no: 02/21 AG) and was carried out according to the Declaration of Helsinki. All consecutive patients, 18 years and older, without signs and symptoms of infection, who were admitted for treatment with allogeneic HSCT and provided written informed consent, were included in the study. Allogeneic HSCT was performed following the Joint Accreditation Committee of the International Society for Cell and Gene Therapy (ISCT) and the European Society for Blood and Marrow Transplantation (EBMT) (JACIE) standards [37]. All patients received standard immunosuppression, prophylactic antibiotics, and antiviral and antifungal therapy, according to the local institutional protocols. The laboratory values of investigated biomarkers were not made available to treating physicians. The following patient characteristics were documented: age, gender, prior medical history, underlying disease, hematopoietic cell transplantation-specific comorbidity index, donor type, stem cell source, conditioning regimen, and prophylaxis protocols.

### 2.2. Clinical Events

Fever was defined as oral body temperature ≥ 38 °C measured on two or more occasions (at least two hours apart) or with one measurement ≥ 38.3 °C [38]. A new febrile episode (FE) was defined as a new fever after more than 72 h of apyrexia (body temperature < 38 °C) [39]. Engraftment was defined as the first of three consecutive days with an absolute neutrophil (ANC) count ≥ 0.5 × 10^9^/L [40]. Neutropenia was defined as an ANC < 0.5 × 10^9^/L or if it was expected to fall below 0.5 × 10^9^/L [38]. At the onset of each FE, a thorough physical examination was carried out, and blood cultures were taken (initially and with each subsequent fever) along with other relevant microbiological samples. When the fever persisted over three to five days, antimicrobial treatment was revised, and appropriate additional evaluation (including microbiological samples and radiological and invasive diagnostics) was carried out according to clinical presentation and on the indication of the attending physician. For analytical purposes, FEs were classified into two groups based on the results of clinical, laboratory, radiological, and microbiological assessment (blinded to PSP and YKL-40 values) according to The Immunocompromised Host Society standard definitions of infections [41]. The first group included FEs with a documented infection (either clinically or microbiologically) and the second group included FEs where no infective cause could be identified, and therefore, those were classified as a fever of unknown origin (FUO)**.** Clinically documented infection (CDI) was designated when there were clinical signs and symptoms and/or radiological findings of infection but a microbiological pathogen could not be proven. A microbiologically documented infection (MDI) included either bacteremia or a defined site of infection with or without positive blood cultures. Different infection foci were defined according to The Centers for Disease Control definitions (CDC) [42]. Mucositis with a World Health Organization (WHO) score > 2 and diarrhea with a frequency of more than 8 times a day were considered clinically documented infections [39]. The isolation of a microorganism from a single blood culture was considered significant if the microorganism was thought to be a relevant cause of infection. In contrast, the identification of an opportunistic microorganism (coagulase-negative staphylococci or *Corynebacterium* species) was considered significant if it was isolated from at least two consecutive blood cultures obtained from separate venipunctures draws or if there were signs of an accompanying venous catheter or skin infection [43]. Cytomegalovirus (CMV) infection was defined as a polymerase chain reaction test (PCR) higher than 500 copies/mL [44].

### 2.3. Biomarker Measurements

Blood samples for measuring PSP and YKL-40 levels were collected upon admission (Day 0), representing baseline levels, and subsequently, if the patient became febrile during the pre-engraftment phase, within 24 h of the onset of fever (Day 1) and on the third (Day 3), fifth (Day 5), and seventh days (Day 7). Venous blood was sampled into vacuum tubes with a clot activator and gel separator (Vacuette, Greiner Bio-One GmbH, Kremsmünster, Austria) and immediately transported to the Clinical Department of Laboratory Diagnostics, University Hospital Centre Zagreb. Blood samples were centrifuged and serum samples were separated, aliquoted into secondary tubes, and stored at −80 °C until analysis. The concentration of YKL-40 and PSP was determined on the Tecan Sunrise absorbance reader with Magellan software version 7.3 (Tecan Group A.G., Männedorf, Switzerland). YKL-40 concentration was determined using a Human Chitinase 3-like 1 Immunoassay ELISA kit (ref. DC3L10 R&D Systems Inc., Minneapolis, MN, USA). The manufacturer-claimed linearity is 3.55–4000 pg/mL. The presepsin concentration was determined using the Human Presepsin ELISA Kit (ref. MBS766136, MyBioSource Inc., San Diego, CA, USA). The manufacturer-claimed linearity is 0.156–10 ng/mL. The concentration of PCT was determined on the Abbott Alinity CI system (Abbott Laboratories, Abbot Park, IL, USA). Procalcitonin was measured using an Abbott Alinity and BRAHMS PCT Reagent Kit (ref. 01R1832, Abbott Laboratories, Abbot Park, IL, USA) with an in-house verified linearity of 0.06–100.00 μg/L.

### 2.4. Statistical Analysis

The normality of the distribution of numerical variables was tested using the Shapiro–Wilk test. Qualitative variables are shown as absolute and relative frequencies. Quantitative variables are shown as mean ± standard deviation (SD) or median with corresponding interquartile range (IQR), depending on the distribution. Differences in biomarker values between patient groups were tested using the Mann–Whitney U test. The Friedman test, followed by Dunn’s post hoc pairwise comparison with the Bonferroni correction, was used to assess whether there were significant differences in biomarker levels over time. Receiver-operating characteristic (ROC) curve analyses were performed with the calculation of the area under the curve (AUC) to assess the diagnostic validity for each biomarker at all examined time points. The optimal cut-off values were determined through maximizing the Youden index. Additional diagnostic accuracy measures (sensitivity, specificity, positive and negative likelihood level, diagnostic odds ratio) were calculated. The correlation between variables was examined using Spearman’s rank correlation test. Univariate binary logistic regression was carried out to assess the individual predictive power of biomarker values. Multiple logistic regression models were developed to evaluate the potential combined predictive abilities for infection. This research’s statistical significance level was set at α = 0.05, with adjusted *p*-values for multiple comparisons. Statistical analyses were conducted using GraphPad Prism version 10.2.3 (GraphPad Software Inc., San Diego, CA, USA).

## 3. Results

### 3.1. Patient and FE Characteristics

Eighty-two patients were admitted to our unit for allogeneic HSCT during the study period. Twenty-eight patients were excluded from the study for the following reasons: five refused to consent, three were infected upon admission, fourteen did not meet fever criteria during the pre-engraftment period, and six had missing data. Therefore, 54 patients and 61 FEs were included in the final analysis. The screening process is shown in Appendix A. The two study physicians agreed with reviewing patient data (κ = 1). The baseline characteristics of the patients and FEs are summarized in Table 1 and Table 2. The mean age of the patients was 46.5 years (SD ± 13.3), with a slight female predominance (53.71%) and acute myeloid leukemia as the most common underlying disease in 25 patients (46.30%). The majority received transplants from matched unrelated donors, accounting for 34 patients (62.96%), and nonmyeloablative or reduced-intensity conditioning regimens were applied to 39 patients (72.22%). Infection was found to be the cause of fever in 38 FEs (62.39%), MDI in 27 episodes (44.26%), and CDI in 11 episodes (18.04%). The remaining 23 FEs (37.70%) were classified as FUO. The majority of infections were caused by Gram-negative bacteria, identified in fourteen episodes (51.85%), followed by viral infections in five episodes (18.52%), Gram-positive infections in four episodes (14.81%), fungal infection in one episode (3.70%), and three episodes that were polymicrobial (11.11%). Gram-positive isolates included *Enterococcus faecium* (*n* = 4), *Enterococcus faecalis* (*n* = 1), *Enterococcus casseliflavus* (*n* = 1), and *Staphylococcus epidermidis* (*n* = 1). Gram-negative isolates included *Pseudomonas aeruginosa* (*n* = 6), *Klebsiella pneumoniae* (*n* = 6), *Escherichia coli* (*n* = 2), *Citrobacter freundii* (*n* = 2), and *Proteus mirabilis* (*n* = 1). The only two fungal findings were *Candida glabrata*. There were five CMV reactivations and one Epstein–Barr virus (EBV) infection. Three confirmed cases of severe acute respiratory syndrome coronavirus 2 (SARSCoV-2) infection were transferred to another department and subsequently lost from follow-up and thus were not included in the final analysis. Out of all the isolated pathogens, 11 (40.74%) were multidrug-resistant organisms. The gastrointestinal tract emerged as the most common source of infection, accounting for 21 episodes (55.26%). No infections originated from skin and soft tissues; the infection source remained unknown in 11 episodes (28.95%). Additionally, bacteremia was detected in 10 episodes (16.39%). Most episodes, 44 of them (72.13%), occurred during the neutropenic period. In total, 7 out of 61 FEs (11.48%) had a quick Sepsis-Related Organ Failure Assessment (qSOFA) score of 2 or more, indicating a greater risk of poor outcome. Unfortunately, one patient died from septic shock eight days after the beginning of the conditioning regimen.

### 3.2. Biomarkers—Diagnostic Potential

#### 3.2.1. Infection vs. FUO

At baseline (Day 0), no significant differences in PSP and YKL-40 concentrations between the infection and FUO groups were observed (1.442 ng/mL vs. 1.220 ng/mL, *p* = 0.483 and 37.60 ng/mL vs. 26.90 ng/mL, *p* = 0.280). All PCT concentrations at baseline were below the detection limit of <0.60 μg/L, so no statistical analysis was performed. On Days 1 and 3, median PSP values were significantly higher in the infection group compared to the FUO (4.835 ng/mL vs. 2.060 ng/mL, *p* = 0.042 and 5.160 ng/mL vs. 2.403 ng/mL, *p* = 0.009). Although the PSP levels were also higher in the infection group on Days 5 and 7 (2.680 ng/mL vs. 2.168 ng/mL and 2.207 ng/mL vs. 1.614 ng/mL), these differences were not significant (*p* = 0.072 and *p* = 0.174). YKL-40 levels did not differ significantly between the infection and FUO groups at any time. PCT concentrations were significantly higher in the infection group compared to the FUO group on Days 3, 5, and 7 (0.45 μg/L vs. 0.17 μg/L, *p* = 0.005; 0.23 μg/L vs. 0.13 μg/L, *p* = 0.027 and 0.19 μg/L vs. 0.09 μg/L, *p* = 0.009, respectively). The value trended to be higher on Day 1 but was not significantly different (0.46 μg/L vs. 0.23 μg/L, *p* = 0.217). The highest median values for biomarkers were observed on Day 3 for PSP and YKL-40 and Day 1 for PCT (Table 3 and Figure 1).

#### 3.2.2. Biomarkers across Time Points

Significant differences in PSP, YKL-40, and PCT concentrations were observed across the investigated time points (Days 0, 1, 3, 5, and 7) in both infection and FUO groups (*p* < 0.001 for all groups, Friedman test) (Figure 2, Appendix A). PSP concentrations increased significantly from baseline (Day 0) to Day 1 and Day 3 in both groups (*p* < 0.001 and *p* = 0.013 in FUO and *p* < 0.001 and *p* < 0.001 in the infection group), with the infection group showing a continued rise by Day 5. This suggests that PSP elevation lasts longer in the infection group compared to FUO. Significant increases in YKL-40 concentrations were observed from baseline to Days 1, 3, 5, and 7 in the FUO group (*p* = 0.001, *p* < 0.001, *p* < 0.001 and *p* = 0.028, respectively). Significant increases in the infection group were found only from baseline to Days 1 and 3 (*p* < 0.001 for both). However, these increases within each group did not translate into significant differences between the FUO and infection groups in YKL-40 concentrations. PCT concentrations increased significantly from baseline to Days 1, 3, and 5 in the FUO group (*p* < 0.001 for Days 1 and 3, and *p* = 0.003 for Day 5) and from baseline through Day 7 in the infection group (*p* < 0.001 for Days 1, 3 and 5, and *p* = 0.029 for Day 7). The sustained increase in PCT values over seven days and the significant differences between groups from Days 3 to 7 suggest that PCT is more effective than PSP for differentiating between infectious and non-infectious complications over a more extended period.

When comparing concentrations on Day 1 to those on Days 3, 5, and 7, no significant differences were found between Day 1 and Day 3 for any biomarker in both groups, indicating that biomarkers likely reached a steady state between these time points (Figure 2, Appendix A). In the infection group, PSP and YKL-40 values on Day 5 were significantly lower compared to Day 1 (*p* = 0.033 and *p* = 0.046), and PSP and PCT values on Day 7 were significantly lower than on Day 1 (*p* < 0.001 for both). In the FUO group, significant differences were observed between Day 1 and Day 7 for PSP and PCT (*p* = 0.045 and *p* < 0.001) but not for other time points.

#### 3.2.3. Associations between PSP, YKL-40, and PCT

At baseline, we did not observe any significant correlations between biomarker concentrations (Appendix A). PSP baseline values did not correlate with PCT (r = −0.09, *p* = 0.532), nor did YKL-40 (r = −0.19, *p* = 0.174). PSP did show a weak correlation with YKL-40 baseline concentrations (r = 0.36, *p* = 0.008). After the onset of fever, PSP concentration strongly correlated with PCT concentrations at all measurement points (r = 0.67, r = 0.78, r = 0.71, r = 0.52 with each *p* < 0.001, from Day 1 to Day 7, respectively). YKL-40 concentrations were weak to moderately correlated with both PSP (r = 0.51, *p* < 0.001, r = 0.36, *p* = 0.005, r = 0.06, *p* = 0.652, r = 0.28, *p* = 0.029, respectively) and PCT levels (r = 0.45, *p* < 0.001, r = 0.38, *p* = 0.002, r = 0.26, *p* = 0.05, r = 0.36, *p* = 0.004, respectively) at the same time points. Additionally, we aimed to test the hypothesis that leukocytes are one of the primary sources of PSP, YKL-40, and PCT, as well as the possibility that anti-thymocyte globulin (ATG) treatment could induce biomarker production. Therefore, regardless of infection status, we primarily analyzed their correlation on Day 1. Results are shown in Table 4. Overall, we did not find any significant correlation between YKL-40 level and leukocyte number during the first 24 h after the onset of fever (r = 0.13, *p* = 0.307). PSP and PTC showed weak positive correlations indicating a closer association with leukocyte count during an inflammatory response (r = 0.38, *p* = 0.002 and r = 0.40, *p* = 0.001). ATG treatment showed a strong positive correlation with PSP levels (r = 0.66, *p* < 0.001) and PCT levels (r = 0.63, *p* < 0.001), suggesting a significant relationship. A moderate positive correlation was found between YKL-40 levels and ATG therapy (r = 0.41, *p* = 0.001), signaling a less notable ATG impact.

### 3.3. Biomarkers—Diagnostic Utility

To test the discriminatory capacity of PSP, YKL-40, and PCT for distinguishing infectious from non-infectious causes of fever, we used the ROC curve analysis of biomarker concentrations focusing on post-baseline time points (Day 1 to 7), as shown in Figure 3. The results are summarized in Table 5. Presepsin showed fair diagnostic performance on Day 1 and Day 3 but no significant diagnostic value on Day 5 and Day 7, where we also found unacceptably low sensitivity. With an AUC of 0.656 (95% CI: 0.510–0.802), PSP could identify infection using the optimal cut-off value of 2.910 ng/mL with 68.42% sensitivity and 65.22% specificity on Day 1. The highest AUC (approaching acceptable discriminatory power), 0.698 (95% CI: 0.559–0.837), with a cut-off value of 3.144 ng/mL, a sensitivity of 71.05%, and a specificity of 69.57%, was found on Day 3. YKL-40 generally showed poor diagnostic performance across all examined time points, with all AUC *p*-values greater than 0.05. The highest AUC for YKL-40 was 0.613 (95% CI: 0.468–0.758) on Day 1. PCT yielded the highest AUC values from all examined biomarkers, notably on Day 3 and Day 7. The best diagnostic performance was observed on Day 3 with an AUC of 0.712 (95% CI: 0.572–0.852), an optimal cut-off of 0.28 μg/L, a sensitivity of 71.05%, and a specificity of 73.91%. Given that the measurements of investigated biomarkers showed the highest AUC values on Day 1 and Day 3, further analysis was carried out only for these time points where the diagnostic AUC *p*-value of at least one biomarker was <0.05. We thought this approach was also justified from the clinical point of view, as relevant decisions regarding treating FEs are typically made within the first 24 h and on the third day.

To estimate the individual predictive power of the observed biomarkers for identifying infection episodes, given the optimal cut-off values determined using ROC curve analysis, we performed univariate binary logistic regression using dichotomized biomarker values (higher or lower than cut-off). The results are shown in Appendix A. Higher PSP concentrations demonstrated significant predictive value on Day 1 and Day 3. Patients with PSP levels exceeding 2.910 ng/mL on Day 1 were approximately four times more likely to have an infection (OR 4.06; 95% CI:1.39–12.70; *p* = 0.012). On Day 3, PSP levels above 3.144 ng/mL increased infection likelihood fivefold (OR 5.61; 95% CI: 1.88–18.38; *p* = 0.003). While increased YKL-40 indicated higher odds for infection; the results were not statistically significant either on Day1 (OR 2.91; 95% CI: 0.94–10.30; *p* = 0.075) or on Day 3 (OR 3.46; 95% CI: 1.05–13.68; *p* = 0.053). PCT levels above 0.28 μg/L on Day 3 strongly predicted infectious episodes (OR 7.93; 95% CI: 2.56–27.66, *p* < 0.001), indicating an almost eightfold increased probability.

Furthermore, several multiple logistic regression models were developed, combining different biomarker levels to adjust for potential interactions between them (Appendix A). Anti-thymocyte globulin therapy was included in models 3 and 6 for potential confounding adjustment. Combining biomarkers improved predictive accuracy. Model 5 on Day 3, combining PSP > 3.144 ng/mL and PCT > 0.28 μg/L, was deemed most acceptable from a statistical and clinical point of view (*p* < 0.001 and R^2^ = 0.3102), with PCT > 0.28 μg/L being the only independent predictor (OR 4.998, 95% CI: 1.380–19.610, *p* = 0.016). As for Day 1, model 2 with two variables (PSP > 2.910 ng/mL and PCT > 0.57 μg/L) was the most reliable choice with *p* = 0.027 and R^2^ = 0.1528, and PSP > 2.910 ng/mL as the only significant predictor (OR 3.463, 95% CI: 1.092–11.520, *p* = 0.038) (Figure 4).

## 4. Discussion

In this prospective study, we assessed the diagnostic relevance of PSP and YKL-40 in discriminating infectious from non-infectious causes of fever during the pre-engraftment phase of allogeneic HSCT. Additionally, we compared their diagnostic performance to that of PCT. Our results showed that PSP had fair discriminatory capacity on Day 1 and Day 3 but not on later days after fever onset. Higher PSP values moderately predicted infection, with approximately four times higher probability on Day 1 and five times higher on Day 3 if above the optimal cut-off values. Compared to YKL-40, PSP had superior diagnostic performance across all time points, but compared to PCT, PSP appeared to be a slightly more effective predictor only on Day 1. This advantage should be interpreted cautiously, considering the wide confidence interval of the PSP AUC (95% CI: 0.510–0.802) and *p*-value of 0.04, highlighting the uncertainty in this estimate. On Day 3, PCT had the best diagnostic performance among all three biomarkers, with values above 0.28 μg/L showing the strongest association with infection. Including PCT to PSP on Day 1 modestly improved the diagnostic performance of PSP, with the combined model explaining approximately 5% more variance (R^2^ increases from about 10.3% to 15.3%) compared to using either biomarker alone. Adding PCT to PSP on Day 3 resulted in an additional 11.2% of the variance explained (R^2^ increased from approximately 20% to 31.2%) and significantly improved explanatory power compared to PCT or PSP values alone.

Data on the PSP value in diagnosing infections in adult hematology patients are scarce, and several small studies have shown variable results. A study by Ebihara et al. focused on evaluating the effectiveness of PSP, C-reactive protein (CRP), and PCT (measured within 72 h of the onset of fever) as infection markers in 49 FEs in 28 patients post-chemotherapy or HSCT [31]. In line with our results, they identified PCT as the best biomarker for infection within 72 h of fever onset, reporting slightly higher diagnostic accuracy than we did. PSP concentrations did not show significant differences between infected and uninfected patients, although they tended to be higher in the infection group. This is contrary to our findings of PSP having the potential to be both an early (Day 1) and later (Day 3) predictor of infection during FEs. However, its predictive capacity seems to be modest. The authors speculated that the lack of difference in PSP levels could be due to a lower monocyte number and decreased phagocytic activity in neutropenic patients, which could result in diminished PSP synthesis in infection. Despite a high proportion of neutropenic FEs (72%), we could still detect significant differences in PSP values between the infected and uninfected groups, including a neutropenic subgroup at multiple time points (Appendix A). Our study also found a weak but statistically significant positive correlation between leukocyte count and PSP levels (r = 0.38, *p* = 0.002). This suggests that although we noticed some association between leukocyte count and PSP concentrations, leukopenia (typically accompanied by neutropenia) might not have a crucial impact on PSP level. Instead, other mechanisms that could induce PSP synthesis might outweigh the negative influence of leukopenia. This is further supported by the fact that we did not find any statistically significant differences in biomarker concentrations between infection and FUO groups within the non-neutropenic patient subgroup, although the small sample size should be taken into account (Appendix A). It is known that chemotherapy, including myeloablative, significantly reduces leukocyte counts. However, some circulating monocytes and tissue-resident macrophages (which are capable of local self-renewal independent of hematopoiesis) survive and can exhibit increased compensatory activity to maintain an immune response even in leukopenia conditions, potentially giving a plausible explanation for higher PSP levels in the infection group [45,46]. Consistent with our findings, many studies have shown neutropenic patients can generate an adequate inflammatory response [33,35,47,48,49].

Kostić et al. assessed the importance of PSP, PCT, IL-8, and CRP (90 to 120 min from the fever onset) in predicting bacteremia in 36 febrile neutropenic episodes in 28 hemato-oncological patients [33]. Opposite to our findings, they found that PCT was the most accurate biomarker for predicting bacteremia, with a cut-off of 1.6 μg/L and an AUC of 0.90. PSP followed it closely with a 410 pg/mL cut-off and an AUC of 0.85.

For patients undergoing HSCT, we found only one study by Stoma et al. that examined the importance of PSP together with PCT and CRP (measured within four hours after fever onset) as a marker of Gram-negative bacteremia in 52 neutropenic patients who underwent autologous or allogeneic HSCT [36]. In agreement with our results, they found that PSP was a more valuable diagnostic marker on the first day after fever onset compared to PCT, with an AUC value for PSP of 0.889 and an optimal cut-off of 218 pg/mL; for PCT, an AUC of 0.741 cut-off value of 1.5 ng/mL was found. Regarding overall PSP diagnostic performance on Day 1 in the hematologic patient population, we generally found it inferior to the findings that Stoma et al. and Kostić et al. reported.

A recent meta-analysis by Kondo et al., including 19 studies and 3012 critically ill adult, non-hematologic patients, found that PSP and PCT had much better diagnostic accuracy [50]. The authors concluded that PSP and PCT could be reliable early biomarkers for diagnosing sepsis, highlighting their compatible diagnostic value. The results were consistent with an earlier similar meta-analysis by Wu et al. [51] but contrasted with our findings, which showed much worse PSP diagnostic performance.

These discrepancies could be due to several hypothetical reasons. The aforementioned studies primarily focused on bacteremia and sepsis as more severe, invasive forms of infection, which theoretically could produce higher biomarker concentration and yield higher AUCs, as it has been shown that PCT and PSP levels positively correlate with infection severity [17,52,53]. In contrast, our study did not specifically target severe forms and included a broader range of infections, likely resulting in greater variability in biomarker production and possibly lower diagnostic performance. This is further supported by the fact that our results are most comparable to the study by Ebihara et al., which provided a population and clinical context similar to our research. Furthermore, the timing of the sample collection could also impact biomarker concentrations and diagnostic validity. It is known that PSP levels typically increase within 2 h and peak at 3 h, while PCT levels increase within 8 to 24 h and peak later than 24 h after infection [13]. We allowed sample collection within a time frame of up to 24 h after fever onset, which might have caused us to miss the initial significant rise and peak in PSP levels at the beginning of the infectious episode. This could explain the diagnostic underperformance we observed for PSP on Day 1 compared to other studies that specified earlier sample timing. These biomarker time dynamics could also explain why PCT exhibited its maximal diagnostic potential later on Day 3. The activation of non-specific inflammatory response, including PSP and PCT-producing macrophages and monocytes, has been observed in patients early after allogeneic stem cell transplant infusion and in those treated with ATG due to cytokine release syndrome [54]. This could increase biomarker concentrations in patients without infection, potentially having a negative impact on the diagnostic performance of all three biomarkers.

It is noteworthy that our study found higher median and cut-off PSP values than the studies mentioned above and previous reports [16,17]. This difference could be due to our specific study population and treatment, but it is more likely attributed to the different laboratory measurement techniques used. We measured PSP concentrations using a human PSP ELISA kit, whereas the earlier studies we discussed in this paper used the more sensitive chemiluminescence enzyme immunoassay (CLEIA)—PATHFAST system. Due to the nature of immunochemistry methods, which utilize different reagent antibodies targeting various parts of the PSP molecule, differences in median and cut-off values between assays are expected. ELISA is a widely used, cost-effective method in research settings, while the more automated PATHFAST system offers a broader assay range but requires more specialized equipment. Okamura and Yokoi, the developers of PATHFAST presepsin assay, showed a strong correlation between their assay and two-step ELISA (r = 0.984), indicating that results from both assays reflect changes in presepsin levels consistently. However, the regression slope of 0.00727 suggests that absolute values might differ between two tests, with ELISA assay concentrations expected to be higher [55]. This methodological difference makes any direct comparison of the PSP absolute concentrations from the two tests difficult. A more thorough investigation, where both methods would be run in parallel, is needed to determine which method is a superior diagnostic tool. Our results were more comparable when comparing median concentrations with other studies that used PSP ELISA kits for diagnosing different infections [56,57,58]. The most recent study by Moustafa et al., which also used an ELISA kit for PSP measurement, investigated diagnostic roles of PCT, PSP, and CRP for identifying infectious complications on the first and third day after the onset of febrile neutropenia in 60 patients treated for acute leukemia [35]. They reported that PCT was a highly accurate infection biomarker on both days, with an AUC of 0.931 on the first and 0.813 on the third. PSP showed worse performance when compared to PCT. They reported an AUC of 0.766 and 0.641 and cut-off values of 1.75 μg/L (1.75 ng/mL) and 2.9 μg/L (2.9 ng/mL) for the first and third days, respectively. Interestingly, both the AUC and cut-off values they found aligned with our findings.

We showed that YKL-40 concentrations significantly increased from the baseline levels in response to an inflammatory stimulus, as was previously reported [22,28,59]. However, the degree of increase did not significantly differ between the two groups, and YKL-40 did not provide any diagnostic value for distinguishing infectious and non-infectious causes of fever, having inferior diagnostic performance compared to PSP and PCT. ROC curve analysis yielded very low AUC values and clinically unacceptable low sensitivity across all measured time points.

Serum YKL-40 levels were comparable to those reported in healthy controls at baseline [22,60,61]. After fever onset, the YKL-40 values we found were, on average, similar to those reported in patients with non-infectious inflammatory conditions, such as rheumatoid arthritis [59], asthma [62], and chronic obstructive pulmonary disease (COPD) [63]. However, they were significantly lower than those previously found in severe bacterial infections, such as severe sepsis, septic shock [27], and streptococcal bacteremia [22], where values were as high as 1000 ng/mL, a threshold we did not observe throughout our study at any point.

It has been shown that the synthesis of YKL-40 increases during infection and that it plays a role in the host’s immune response against bacterial infections [21,25]. An increased expression of YKL-40 has also been observed in non-infectious inflammatory conditions such as rheumatoid arthritis [64] and COPD [63]. It is produced in various connective tissue cells [65], tumor cells [66], activated neutrophils [67], and macrophages [68]. Inflammatory cytokines regulate its formation. In activated macrophages, the synthesis is stimulated by interferon-γ and is inhibited by IL-4 [68]. In healthy volunteers, YKL-40 values in the serum were stimulated via the administration of recombinant human IL-6 [69].

The entire procedure of allogeneic HSCT represents a state characterized by increased inflammatory activity, which could have resulted in the higher YKL-40 concentrations observed in the present study. YKL-40 also benefits the migration and proliferation of vascular endothelial and smooth muscle cells in response to tissue injury, indicating its possible role in tissue remodeling and repair during the inflammatory process [70,71]. This could explain the steadily elevated levels observed in our study, in addition to the ongoing inflammation. The inability to distinguish between infectious and non-infectious FEs was likely due to the less severe nature of infectious episodes in our study, as well as the fact that YKL-40 concentrations most likely reflect the overall intensity and severity of the background inflammatory response in the body, rather than explicitly detecting the infectious pathogen itself, as previously hypothesized by Kornbilt [72].

We did not find any difference in the investigated biomarker concentrations depending on the type of isolated pathogen (Gram-positive, Gram-negative, fungal, viral or polymicrobial), nor between multidrug-resistant and non-multidrug-resistant microorganisms (Appendix A). However, it is important to note that the lack of significant difference may be due to the small sample sizes in each group. Larger studies are needed to investigate potential differences in PSP and YKL-40 concentrations between different infection types.

Our study had several limitations. It was a single-center study that included a small number of FEs and patients (although comparable to most previous single-center studies), which may affect the generalizability of the results. Furthermore, the flexible initial timeframe for biomarker sampling might have failed to capture the full extent of the potential biomarker diagnostic effect of the investigated biomarkers. Additionally, differences in laboratory measurement techniques could limit the reproducibility and comparability of our results. We need larger, multicenter studies to validate our results and better understand the role of PSP and YKL-40 in patients undergoing allogeneic HSCT.

## 5. Conclusions

PSP showed fair diagnostic value on Day 1 and Day 3 for identifying infections in febrile patients during the pre-engraftment phase of allogeneic HSCT. YKL-40 showed poor diagnostic performance across all examined time points. PCT outperformed both PSP and YKL-40 in diagnostic accuracy, particularly on Day 3, and seems more useful as an infection biomarker in the later stages of fever. Multivariable logistic models suggest that combining PSP with PCT can provide better predictive power than either biomarker alone. However, neither of these tests showed acceptable discriminative power as an early infection biomarker, and we cannot recommend using any of them alone in everyday clinical practice. At the same time, we wish to point out the potential benefit of a multimodal approach combining multiple biomarkers to improve diagnostic accuracy.

## Figures and Tables

**Figure 1 jcm-13-05991-f001:**
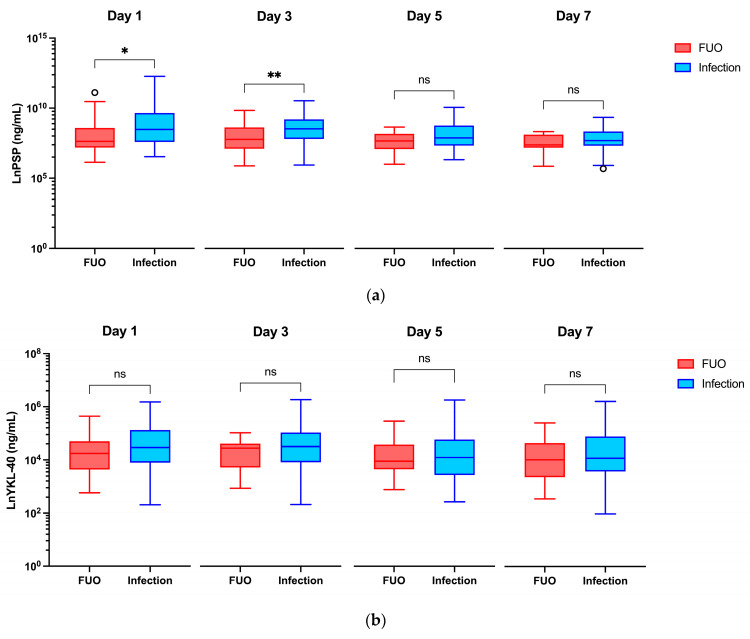
Serum values of PSP, YKL-40, and PCT during the first seven days after the onset of FE. Values are plotted on a linear scale after natural logarithm transformation due to their wide range, allowing better visualization. (**a**) Differences in median LnPSP concentrations. (**b**) Differences in median LnYKL-40 concentrations. (**c**) Differences in median LnPCT concentrations. Each box-and-whiskers plot (drawn using the Tuckey method) depicts the median value (horizontal line) and the values from the first to third quartile (central box). Each whisker extends from the edges of the box to the smallest and largest values within 1.5 times the interquartile range. Outliers are depicted as circles. Differences in biomarker values between FUO and infection groups were tested using the Mann–Whitney U test. ns = non-significant, * = *p* < 0.05, ** = *p* < 0.01. FE, febrile episode; FUO, fever of unknown origin; PCT, procalcitonin; PSP, presepsin.

**Figure 2 jcm-13-05991-f002:**
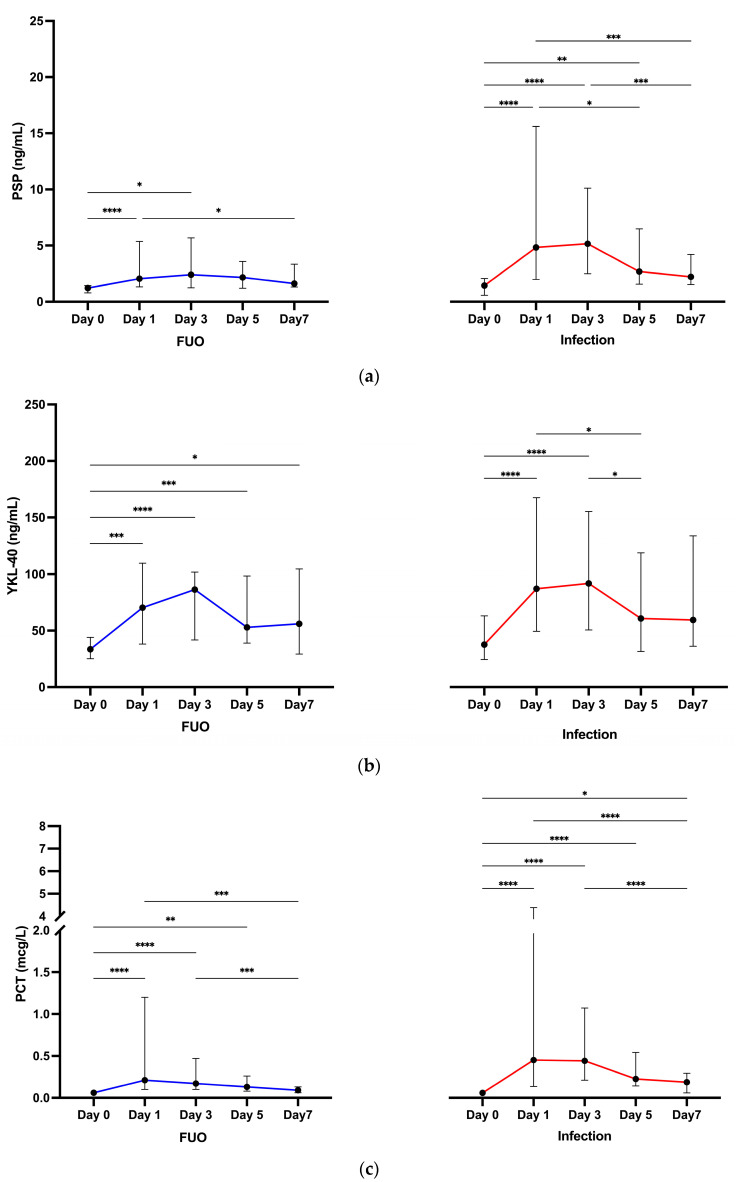
Differences in biomarker median concentrations in FUO and the infection group across Day 0, Day 1, Day 3, Day 5, and Day 7 time points. (**a**) Differences in median PSP concentrations. (**b**) Differences in median YKL-40 concentrations. (**c**) Differences in median PCT concentrations. Results are shown as median values with interquartile range. All values for PCT Day 0 in both groups were below the detection limit (<0.6 μg/L). To ensure a conservative estimate and clinical relevance, for statistical analysis, the values were substituted with the detection limit of 0.6 μg/L. Adjusted *p* value: * = *p* < 0.05, ** = *p* < 0.01, *** = *p* < 0.001, **** = *p* < 0.0001. FUO, fever of unknown origin; PCT, procalcitonin; PSP, presepsin.

**Figure 3 jcm-13-05991-f003:**
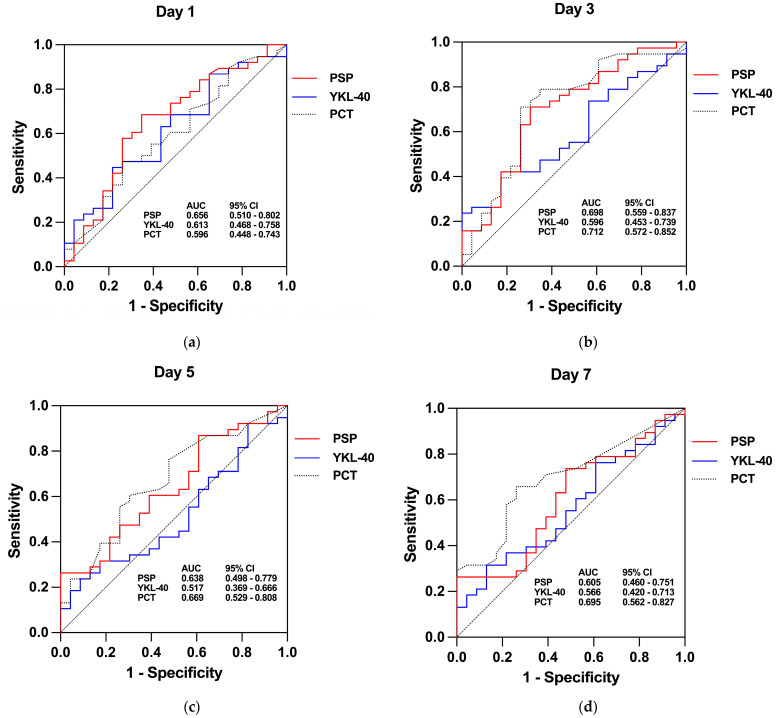
Receiver operating characteristic (ROC) curve analysis data for PSP, YKL-40, and PCT serum concentrations to differentiate infectious from non-infectious causes of fever at (**a**) Day 1, (**b**) Day 3, (**c**) Day 5, and (**d**) Day 7 after the onset of fever. For details on AUC *p*-values, please see Table 5. AUC, area under the ROC curve; PCT, procalcitonin; PSP, presepsin.

**Figure 4 jcm-13-05991-f004:**
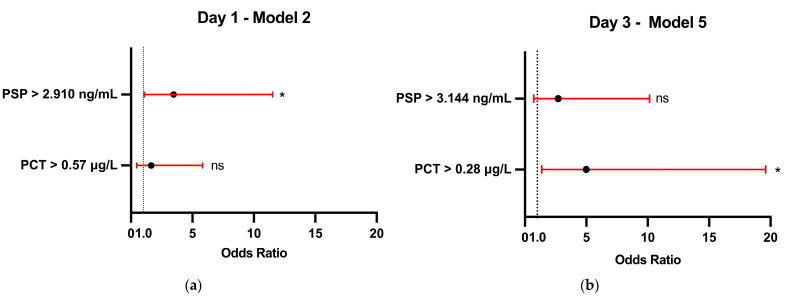
Best multivariable logistic regression models showing odds ratio for infection for PSP and PCT on (**a**) Day 1 and (**b**) Day 3. The black circles represent the estimated odds ratio, and the horizontal red lines with whiskers show the 95% confidence interval. An odds ratio greater than 1 indicates an increased likelihood of infection. * = *p* < 0.05.

**Table 1 jcm-13-05991-t001:** Baseline characteristics of the study population.

Patients	*n* = 54
Age (mean ± SD)	46.50 ± 13.3
Gender, *n* (%)	
Female	29 (53.71)
Male	25 (46.29)
HCT-CI score, *n* (%)	
Low (0)	20 (37.04)
Intermediate (1–2)	23 (42.59)
High (≥3)	11 (20.37)
Underlying disease, *n* (%)	
AML	25 (46.30)
ALL	11 (20.37)
Lymphoma	8 (14.81)
MPN	3 (5.56)
MDS	2 (3.70)
Aplastic anemia	1 (1.85)
Rest	4 (7.41)
Donor type, *n* (%)	
Matched unrelated	34 (62.96)
Haploidentical	12 (22.22)
Matched related	8 (14.81)
Conditioning regimen, *n* (%)	
Nonmyeloablative/RIC	39 (72.22)
Myeloablative	15 (27.78)

AML, acute myeloid leukemia; ALL, Acute lymphoblastic leukemia; HCT-CI, hematopoietic cell transplantation—comorbidity index; MDS, myelodysplastic syndrome; MPN, myeloproliferative neoplasms; RIC, reduced intensity conditioning.

**Table 2 jcm-13-05991-t002:** Baseline characteristics of FEs.

Febrile Episodes (FE)	*n* = 61
Recurrent, *n* (%)	7 (11.48)
Documented infection, *n* (%)	38 (62.30)
MDI	27 (44.26)
CDI	11 (18.04)
FUO, *n* (%)	23 (37.70)
Isolated pathogen, *n* (%)	
G− bacteria	14 (51.85)
Viral	5 (18.52)
G+ bacteria	4 (14.81)
Polymicrobial	3 (11.11)
Fungal	1 (3.70)
Infection source, *n* (%)	
Gastrointestinal	21 (55.26)
Genitourinary	4 (10.53)
Respiratory	2 (5.26)
Skin and soft tissues	0 (0)
Unknown	11 (28.95)
Bacteremia, *n* (%)	10 (16.39)
Neutropenia, *n* (%)	44 (72.13)
Duration of neutropenia before FE (days), median (range)	4 (0–8)
Acute GVHD, *n* (%)	7 (11.48)
Engraftment (days), median (range)	16 (14–21)
ATG, *n* (%)	18 (29.51)
G-CSF, *n* (%)	22 (36.07)
Infection severity, *n* (%)	
qSOFA score ≥ 2	7 (11.48)
Fatal outcome ^a^	1 (1.64)

ATG, anti-thymocyte globulin; CDI, clinically documented infection; FE, febrile episode; FUO, fever of unknown origin; G-, Gram-negative; G-CSF, granulocyte colony-stimulating factor; G+, Gram-positive; GVHD, graft-versus-host disease; MDI, microbiologically documented infection; qSOFA, quick Sepsis-Related Organ Failure Assessment; ^a^ 30-day mortality.

**Table 3 jcm-13-05991-t003:** Serum values of PSP, YKL-40, and PCT at admission (representing baseline values) (Day 0) and during the first seven days after the onset of the FE (Day 1, Day 3, Day 5, Day 7).

Biomarker	Time Point (Day)	Infection (*n* = 38)	FUO (*n* = 23)	*p*-Value
PSP (ng/mL)	Day 0	1.442 (0.560–2.424)	1.220 (0.762–1.560)	0.483
	Day 1	4.835 (1.979–15.600)	2.060 (1.322–5.383)	0.042 *
	Day 3	5.160 (2.492–10.110)	2.403 (1.236–5.686)	0.009 **
	Day 5	2.680 (1.553–6.489)	2.168 (1.204–3.596)	0.072
	Day 7	2.207 (1.524–4.206)	1.614 (1.320–3.354)	0.174
YKL-40 (ng/mL)	Day 0	37.60 (21.80–70.65)	26.90 (21.30–44.90)	0.280
	Day 1	87.10 (49.40–167.60)	70.30 (38.10–109.70)	0.143
	Day 3	91.80 (50.65–155.30)	86.40 (41.80–101.90)	0.215
	Day 5	60.80 (31.60–118.70)	52.90 (39.00–98.40)	0.828
	Day 7	59.40 (36.18–133.70)	56.00 (29.30–104.70)	0.393
PCT (μg/L)	Day 0	N/D	N/D	N/A
	Day 1	0.46 (0.14–4.53)	0.23 (0.10–1.25)	0.217
	Day 3	0.45 (0.22–1.09)	0.17 (0.10–0.47)	0.005 **
	Day 5	0.23 (0.15–0.47)	0.13 (0.08–0.26)	0.027 *
	Day 7	0.19 (0.06–0.30)	0.09 (0.06–0.13)	0.009 **

The data are expressed as medians and interquartile ranges. Differences in biomarker values between the infection and FUO group were tested using the Mann–Whitney U test. All PCT Day 0 values were nondetectable (below the detection limit of <0.6 μg/L). Therefore, no statistical test was performed for PCT Day 0. * = *p* < 0.05, ** = *p* < 0.01. FE, febrile episode; FUO, fever of unknown origin; N/A, not applicable; N/D, nondetectable; PCT, procalcitonin; PSP, presepsin.

**Table 4 jcm-13-05991-t004:** Spearman correlations between serum levels of PSP, YKL-40, PCT, leukocyte count, and ATG therapy on Day 1, independent of infection status.

	Leukocyte	PSP	YKL-40	PCT	ATG Therapy
Leukocyte					
r	1.00	0.38	0.13	0.40	0.50
95% CI		0.14 to 0.58	−0.13 to 0.38	0.16 to 0.60	0.28 to 0.67
*p*		0.002	0.307	0.001	<0.001
PSP					
r		1.00	0.51	0.67	0.66
95% CI			0.29 to 0.68	0.50 to 0.79	0.48 to 0.78
*p*			<0.001	<0.001	<0.001
YKL-40					
r			1.00	0.45	0.41
95% CI				0.22 to 0.64	0.17 to 0.60
*p*				<0.001	0.001
PCT					
r				1.00	0.63
95% CI					0.45 to 0.77
*p*					<0.001
ATG therapy					
r					1.00
95% CI					
*p*					

A strong positive correlation was defined as r > 0.6 and *p* ≤ 0.05. ATG, anti-thymocyte globulin; PCT, procalcitonin; PSP, presepsin; r, Spearman correlation coefficient.

**Table 5 jcm-13-05991-t005:** Diagnostic accuracy criteria of PSP, YKL-40, and PCT serum concentration for identifying infective FEs on Day 1, Day 3, Day 5, and Day 7 after the beginning of the FE.

Biomarker	Day	AUC (95% CI)	Cut-Off	Sensitivity (%)	Specificity (%)	Youden Index	LH+	LH−	DOR	*p* Value
PSP (ng/mL)	Day 1	0.656 (0.510–0.802)	2.910	68.42	65.22	0.34	1.967	0.484	4.063	0.042 *
	Day 3	0.698 (0.559–0.837)	3.144	71.05	69.57	0.41	2.335	0.416	5.612	0.010 *
	Day 5	0.638 (0.498–0.779)	5.585	26.32	95.64	0.22	6.051	0.770	7.855	0.072
	Day 7	0.605 (0.459–0.751)	3.991	26.32	95.65	0.22	6.051	0.770	7.855	0.171
YKL-40 (ng/mL)	Day 1	0.613 (0.468–0.758)	111.40	44.74	78.26	0.23	2.058	0.706	2.915	0.141
	Day 3	0.596 (0.453–0.739)	108.40	42.11	82.61	0.25	2.422	0.701	3.456	0.211
	Day 5	0.517 (0.369–0.666)	31.45	23.68	91.30	0.15	2.722	0.836	3.256	0.823
	Day 7	0.566 (0.420–0.712)	119.80	31.58	86.96	0.19	2.422	0.787	3.078	0.388
PCT (μg/L)	Day 1	0.596 (0.448–0.743)	0.57	47.37	73.91	0.21	1.816	0.712	2.550	0.214
	Day 3	0.712 (0.572–0.852)	0.28	71.05	73.91	0.45	2.723	0.392	6.953	0.006 **
	Day 5	0.669 (0.529–0.808)	0.19	60.53	69.57	0.30	1.989	0.567	3.506	0.028 *
	Day 7	0.694 (0.562–0.827)	0.12	65.79	73.91	0.40	2.522	0.463	5.448	0.011 *

* = *p* < 0.05, ** = *p* < 0.01. AUC, area under the ROC curve; DOR, diagnostic odds ratio; FE, febrile episode; LH –, negative likelihood ratio; LH +, positive likelihood ratio; PCT, procalcitonin; PSP, presepsin.

## Data Availability

The original data presented in the study are openly available at http://www.doi.org/10.17605/OSF.IO/4ZQ9E, accessed on 5 September 2024.

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
