# Peer review of "Early Fever in Allogeneic Stem Cell Transplantation: Are Presepsin and YKL-40 Valuable Diagnostic Tools?"

_jcm, 2024, doi:10.3390/jcm13195991_

Round 1

Reviewer 1 Report

Comments and Suggestions for Authors

The prompt identification of infective complications after stem cell transplantation could increase remains challenging and can increase non relapse mortality. In this paper the authors investigated the discriminative power for infection during the pre-engraftment phase of serum levels of presepsin (PSP) and YKL-40, compared to that of procalcitonin (PCT). 

They concluded that neither test is recommended as individuals diagnostic tool in clinical practice.

Despite its negative conclusion, the paper contains many interesting points and provides suggestions for further investigations. 

I have just few questions:

·      PSP cut- off levels in this paper was higher compared the literature data. The authors attributed the differences to different analytic methods. Did they identified the best one?

·      In the paper the tests were performed during neutropenia in 44% of cases (table2). Did they observed differences in non-neutropenic patients. Could the tests increase their discriminative power in non- neutropenic? If yes, the tests could be used to predict infections versus GVHD after engraftment.

·      Did the authors find differences in PSP levels and Gram-, Gram+ or Fungal infections? Could the lack of correlation due to the small sample size?

Reviewer 2 Report

Comments and Suggestions for Authors

The authors investigate the potential use of PSP and YKL-40 as markers in febrile episode in allo-HSCT setting. The manuscript is interesting and well written. Following some comments:

Please use the acronym FE consistently across the manuscript.

“When comparing concentrations on Day 1 to those on Days 3, 5, and 7, no significant differences were found between Day 1 and Day 3 for any biomarker in both groups, indicating that biomarkers likely reached a steady state between these time points (Figure 3..” In this sentence I think the authors refer to figure 2.

In the subgroup of patients with documented infection, did the authors find differences in the accuracy of the different biomarkers according to the type of bacteria (e.g. MRDO+ vs MDRO-)?

The quality of the figures (in particular S2) should be improved.

Round 2

Reviewer 1 Report

Comments and Suggestions for Authors

I read the revised manuscript. No further comments